# PERSONALEDGER: GENERATING REALISTIC FINANCIAL TRANSACTIONS WITH PERSONA CONDITIONED LLMS AND RULE GROUNDED FEEDBACK

## ABSTRACT

Strict privacy regulations limit access to real transaction data, slowing open research in financial AI. Synthetic data can bridge this gap, but existing generators do not jointly achieve *behavioral diversity* and *logical groundedness*. Rule-driven simulators rely on hand-crafted workflows and shallow stochasticity, which miss the richness of human behavior. Learning-based generators such as GANs capture correlations yet often violate hard financial constraints and still require training on private data. We introduce **PersonaLedger**, a generation engine that uses a large language model conditioned on rich user personas to produce diverse transaction streams, coupled with an expert configurable *programmatic engine* that maintains *correctness*. The LLM and engine interact in a closed loop: after each event, the engine updates the user state, enforces financial rules, and returns a context aware `next_prompt` that guides the LLM toward feasible next actions. With this engine, we create a public dataset of 30 million transactions from 23,000 users and a benchmark suite with two tasks, *illiquidity classification* and *identity theft segmentation*. PersonaLedger offers a realistic, privacy preserving resource that supports rigorous evaluation of forecasting and anomaly detection models. PersonaLedger offers the community a rich, realistic, and privacy preserving resource—complete with code, rules, and generation logs—to accelerate innovation in financial AI and enable rigorous, reproducible evaluation.

## 1 INTRODUCTION

Digital payments and account transfers generate vast streams of transactional events that are central to forecasting, risk management, and personalized financial services. Yet strict privacy regulations and institutional compliance keep most real ledgers out of public view, limiting open and reproducible research. Public synthetic datasets partly address this scarcity (Padhi et al., 2021) and have stimulated progress on tasks such as fraud detection and account forecasting (Zhang et al., 2023). Still, two fronts matter most for high quality synthetic ledgers and remain unmet at scale: *behavioral diversity* and *logical groundedness*.

**Diversity.** Realistic spending emerges from many weak yet interacting signals. Occupation shapes disposable income and work related purchases; schedules modulate the timing of spend; hobbies and social ties steer merchant choices; life stage shifts priorities and baskets. A college student buys textbooks and inexpensive meals; a new parent buys diapers and childcare; a contractor with irregular pay exhibits liquidity swings. Encoding this space with fixed rules is not scalable and collapses to randomness rather than realistic variety. Thus, a generator must compose rich context without enumerating branches. **Groundedness.** At the same time, transactions must remain consistent with a user's circumstances and with accounting rules over time. A car owner makes periodic fuel and maintenance purchases, while a non owner relies on transit. A nine-to-five worker spends differently on weekdays than a retiree with flexible time. Holidays create distinct patterns for those who stay in versus those who shop. Above all, ledgers must respect hard financial constraints: balances update consistently, spending is funded by income over reasonable horizons, and mandatory payments occur by due dates. Generators without explicit state tracking and rule checks drift into infeasible scenarios. The challenge is therefore **dual**: compose diversity while preserving constraints.

**Limits of existing approaches.** Rule-driven simulators (Altman, 2021; Altman et al., 2023) encode hand-crafted workflows and sample with simple randomness. They rarely capture the interplay between occupation, income cadence, family structure, commute and geography, health needs, and leisure preferences that jointly shape when, where, and how much people spend. Enumerating such confounders by hand is not scalable, which leads to limited variety and unrealistic correlations. Learning-based generators such as GANs (Zhao et al., 2025; Liu et al., 2022a) reproduce surface level statistics but typically ignore hard constraints that real ledgers obey. They often fail to maintain consistent balances, schedule mandatory payments, or keep purchases compatible with ownership and calendar context. Further, these models require access to private data for training, which is not feasible for open release. These constraints motivate a different path.

Large language models provide the missing mechanism for broad yet coherent variety. They support *compositional generalization*, combining attributes that never co-appeared in a template, which covers the exponential space of persona mixtures without enumerating branches. Their broad training imbues *world knowledge* about merchants, seasons, and routines, which induces realistic choices without private ledgers. With *fine grained conditioning*, an LLM proposes $x_t \sim \pi_\theta(\cdot \mid p, s_t)$ given a structured persona $p$ and current state $s_t$, while decoding controls yield multiple plausible ledgers.

Despite their strengths, LLMs are not reliable accountants. As sequences grow longer, *rule following degrades*: small local deviations compound into budget drift, overdue bills, double counting, and other reconciliation errors. Exposure bias makes later steps depend on earlier model outputs rather than truth, so inconsistencies accumulate across $s_t$. Free form text rationales also do not guarantee compliance with accounting identities. In practice, unconstrained decoding often violates basic constraints such as , spend $\leq$ income over rolling horizons, timely payment of liabilities, and consistent balance updates. Therefore, an auxiliary mechanism is required to enforce rules, update state, and correct trajectories in flight.

**Approach overview.** We introduce **PersonaLedger**, a closed loop engine that couples a persona conditioned LLM with an expert configurable programmatic system. The engine maintains an explicit user state $s_t$ with balances, income cadence, liabilities, recurring bills, calendar events, and persona attributes. At round $t$, the LLM proposes a candidate transaction $x_t$ given (persona, $s_t$). The engine verifies constraints $g_i(s_t, x_t) \leq 0$, applies the deterministic update $s_{t+1} = f(s_t, x_t)$ when valid, and otherwise returns a structured `next_prompt` that pinpoints violations and suggests feasible alternatives. In effect, the LLM supplies compositional diversity, while the engine supplies rule grounded guardrails that prevent long horizon drift.

**Dataset and tasks.** Built on this engine, we generate 30 million transactions for 23,000 users over multi year timelines. To make the resource immediately useful, we release two benchmark tasks aligned with practice: *illiquidity classification*, which flags near term cash shortfalls, and *identity theft segmentation*, which localizes anomalous subsequences consistent with account takeover. We report results from modern time series and event sequence baselines and provide train, validation, and test splits with rule satisfaction metadata and state snapshots. PersonaLedger uses no private customer data. We release code, prompts, rule sets, seeds, and generation logs that record every constraint check and state transition, enabling exact regeneration of the corpus, transparent audit of design decisions, and straightforward extension to new policies and markets.

**Contributions.** Our work makes four contributions:

- **Method.** We introduce a *rule grounded, stateful, multi round* generation framework that is novel in coupling persona conditioned LLM proposals with a *programmatic controller* that enforces machine checkable accounting invariants before any event is accepted. Experts define rules in a simple declarative template (`update`, `next_prompt`), enabling targeted repair when violations occur and *plug and play* expansion to new policies and domains without retraining.

- **Resource.** A public corpus of 30,000,000 transactions across 23,000 users with multi year coverage, rich personas, and complete state snapshots. We provide *statistical verification* of diversity and realism (persona, calendar, and lifecycle patterns with broad within group variance) and *programmatically enforced groundedness* (accounting invariants with recorded rule satisfaction per event), yielding a trustworthy base for modeling.

- **Benchmarking.** A difficult suite with two tasks—*illiquidity classification* and *identity theft segmentation*—that stress long horizon reasoning, calibration under class imbalance, and detection of localized anomalies. The suite is *scalable*: by adjusting personas, rules, and economic context,

we can increase difficulty without retraining and avoid saturation as models improve. We release protocols, strong baselines, and headroom diagnostics to track progress.

- **Reproducibility and ethics.** Released code, prompts, rule sets, seeds, and per step generation logs that record constraint checks and state transitions, enabling exact regeneration, ablation, and audit. The dataset contains no private data and includes documentation of risks and usage guidelines to support responsible research and ethics review.

## 2 PERSONALEDGER DATASET

This section describes our PersonaLedger dataset. We begin in Section 2.1 with an overview of the dataset's properties, including its scale, coverage, and examples that show how personas are reflected in the generated transactions. Section 2.2 then presents spending statistics across various dimensions to demonstrate the dataset's diversity and realism. In Section 2.3, we detail the motivation and the generation engine, explaining how data is grounded in a set of refinable constraints.

### 2.1 OVERVIEW

**Dataset overview.** The PersonaLedger dataset pairs augmented *user personas* from Nemotron-Personas (Meyer & Corneil, 2025) with corresponding *transaction and payment sequences*. Each persona is a 20-item dictionary covering the user's demographics, lifestyle, professional background, and financial status. Transaction events are defined by five fields: timestamp, merchant_name, merchant_type, card_present_or_not, and amount. The amount is positive for a transaction and negative for a payment, unifying both event types into a single sequence. See Figure 1 for a partial example and Appendix A for a complete list of fields and full examples.

| user_id = 0 | | user_id | timestamp | merchant_name | merchant_type | card_present_or_not | amount |
|---|---|---|---|---|---|---|---|
| "persona": | "A disciplined, sociable visionary, Jonathan ... ", | 0 | 2024-04-08 09:00:00 | Speedway | Gas | yes | 25.92 |
| "professional_persona": | "A retired manufacturing manager, Jonathan ... ", | 0 | 2024-04-08 10:30:00 | Walmart | grocery | yes | 68.15 |
| "sports_persona": | "An avid golfer, Jonathan plays weekly at ... ", | 0 | 2024-04-09 12:00:00 | Tasty Bites | restaurant | yes | 24.02 |
| ... | ... | ... | ... | ... | ... | ... | ... |
| "income_level": | "med income", | 22017 | 2026-10-31 14:00:00 | Pharmacy | pharmacy | yes | 20.75 |
| "credit_limit": | "9500", | 22017 | 2026-11-01 10:00:00 | Church Donation | charity | yes | 20.00 |
| "spending_habit": | "Balancers" | 22017 | 2026-11-01 19:00:00 | payment thank you | payment | no | -3500.00 |

Figure 1: Our PersonaLedger dataset consists of user personas (Left) and their corresponding transactions and payments sequences (Right).

**Scale and Coverage.** Our dataset contains 30 million transactions from over 23,000 users, with an average activity period of two years per user. The dataset features a diverse range of nearly 75,000 unique merchants from various sectors, including retail, grocery, insurance, etc.. This breadth is complemented by depth, with 809 merchants appearing more than 1,000 times. Furthermore, the dataset is highly **scalable**, as its coverage can be efficiently expanded by adding new user personas. Table 1 summarizes key statistics on the scale and coverage of PersonaLedger, while Section 2.2 and Appendix B provides a more comprehensive exploratory data analysis, including spending distributions, temporal patterns, credit utilization, and payment patterns

Table 1: Statistics of PersonaLedger. The illiquid user in the table is someone whose credit balance exceeds their available cash. These users are the foundation for a benchmarking task in Section 3.

| | | count | | mean | std | min | 25% | 50% | 75% | max |
|---|---|---|---|---|---|---|---|---|---|---|
| events | daily transactions | 24.7M (79.9%) | timespan per user (days) | 724.05 | 366.11 | 89 | 367 | 792 | 1094 | 1101 |
| | payments | 1.9M (6.2%) | #transactions per user | 1242.10 | 642.16 | 127 | 629 | 1347 | 1824 | 4024 |
| | recurring subscriptions | 4.30M (13.9%) | #transactions per month | 49.97 | 12.14 | 1 | 44 | 50 | 57 | 186 |
| merchant names | total | 74623 | transaction amounts | 66.24 | 184.46 | 0.00 | 15.22 | 25.67 | 50.35 | 32696.64 |
| | freq. >1000 | 809 | transaction amounts per month | 3309.69 | 1609.13 | 0.69 | 2257.34 | 3024.83 | 4107.03 | 126576.50 |
| users | normal users | 22018 (94.3%) | payment amounts | 978.74 | 1189.40 | 0.01 | 200.00 | 500.00 | 1434.13 | 34703.13 |
| | illiquid users | 1343 (5.7%) | payment amounts per month | 3293.89 | 1942.03 | 0.32 | 1991.53 | 2993.92 | 4279.98 | 120793.46 |

Figure 2 shows an example that demonstrate **personas and generated transactions are strongly bounded.** In this example, each transaction is a direct result of the user's specific traits and circumstances: the user's sociable nature lets him meet up with friends at a local sports bar, while the fact that he does not own a car causes him to get a ride there, generating an Uber transaction. Later, his desire for a relaxed holiday dinner leads him to order a pizza online, while his career motivation for the new year causes him to purchase a subscription to a Spanish language tool. Every purchase is a

logical footprint of his story, showing how the user persona and the transactions are tightly linked. Consequently, the diverse personas ensure the generated transactions to be diverse.

**trajectory plan:**

As it's New Year's Day, Michael enjoys a slow start, waking up late and having a simple coffee and toast at home. In the afternoon, embracing his sociable nature, he meets up with friends at a local sports bar to watch a college football bowl game, getting a ride there since he doesn't own a car. For lunch, he grabs a burger and a soda at the bar. He heads home in the late afternoon. For a relaxed holiday dinner, he decides to order a pizza online. In the evening, feeling motivated by the new year, he purchases a subscription to a Spanish language learning tool to support his career goal of becoming a medical interpreter.

**generated transactions:**

| timestamp | merchant_name | merchant_type | card_present_or_not | amount |
|---|---|---|---|---|
| 2025-01-01 00:00:00 | Netflix | Streaming service | no | 15.99 |
| 2025-01-01 13:40:15 | Uber | travel | no | 11.52 |
| 2025-01-01 16:15:48 | Grindstone Tap House | dining | yes | 28.74 |
| 2025-01-01 19:05:21 | Domino's Pizza | dining | no | 23.49 |
| 2025-01-01 20:55:10 | Babbel | shopping | no | 14.95 |

Figure 2: Personas and generated transactions are strongly bounded.

## 2.2 DIVERSITY AND REALISM VIEWED FROM STATISTICS

**The dataset is statistically diverse and socioeconomically realistic.** Figure 3 reports average monthly spending grouped by **persona attributes**. The corpus spans seven education levels, both car ownership statuses, and the full adult age range. Patterns align with established regularities: spending increases with education and with car ownership; the age profile follows the classic lifecycle spending curve (Foster, 2015), peaking in middle age and tapering thereafter; and spending rises monotonically from the extremely frugal "survivor" to the high spending "spender". Large error bars indicate substantial within-group heterogeneity, which is expected in real populations.

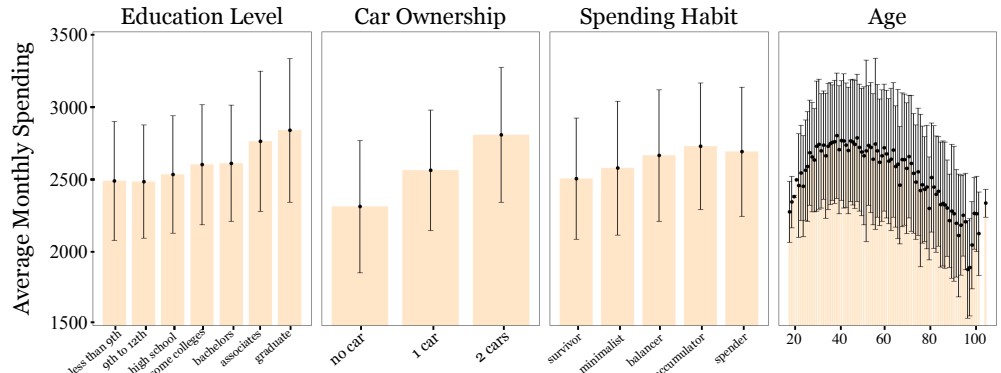

Figure 3: Average monthly spending and error bars grouped by different persona attributes.

**Calendar effects match consumer behavior and variance is larger on holidays.** Figure 4 groups spending by **temporal features**. December spending is modestly higher, consistent with holiday shopping. The increase is muted because the model receives only calendar cues (e.g. "Today is Christmas") rather than explicit promotions. Weekday patterns show slightly elevated spending on Fridays and Saturdays. Average spend is similar between holidays and non-holidays, but variance is markedly larger on holidays, reflecting heterogeneity across dates such as home-oriented observances like Presidents' Day versus commerce heavy occasions like Christmas (U.S. Bureau, 2024).

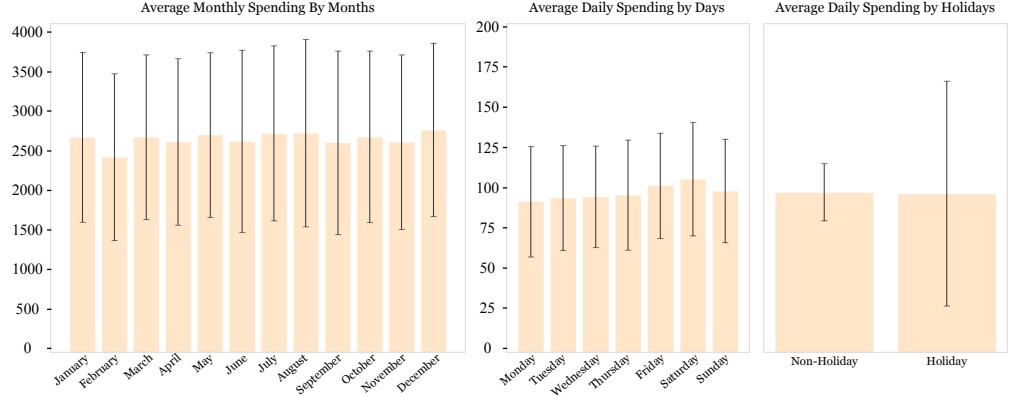

Figure 4: Average monthly/daily spending and error bars grouped by different temporal features.

**Illiquidity manifests as high volatility in credit utilization over time.** Figure 7 plots the **credit utilization rate** (cumulative balance divided by limit) across one year. Users categorized as illiquid display substantially more volatile trajectories than solvent users, consistent with irregular spending and delayed payments near binding constraints.

**Merchant coverage is broad and supports behavioral diversity.** Figure 8 visualizes merchant names in a word cloud, illustrating wide coverage across retail, services, transportation, and lifestyle categories. Taken together, these statistics indicate that the dataset is both *diverse* across personas and calendar contexts and *realistic* in its macro level regularities and variances. We next introduce the rule grounded engine that produces these sequences.

## 2.3 RULE GROUNDED GENERATION

**Motivating rule grounded, multi-round generation.** A naive prompting baseline with Llama-3.3-70B (Figure 5) exposes systematic *accounting* failures: **(A)** unrealistically frequent grocery purchases that ignore prior cadence; **(B)** subscriptions generated but not carried into the next billing cycle; **(C)** an overpayment caused by a miscomputed balance. We further observe missing capital expenditures, muted weekend and holiday effects, and the absence of naturally emergent illiquidity. Critically, these errors accumulate with sequence length, degrading rule following and violating basic reconciliation. This motivates a stateful controller that *enforces accounting correctness* at every step.

Figure 5: A case study of unrealistic transactions from a naive prompting baseline with Llama-3.3-70B.

| timestamp | merchant_name | merchant_type | card_present_or_not | amount |
|---|---|---|---|---|
| 2025-07-01 08:15:23 | Artisan Cafe | cafe | yes | 5.75 |
| 2025-07-01 18:00:10 | Publix Super Markets | grocery | yes | 184.32 |
| 2025-07-01 12:45:22 | Jersey Mike's Subs | restaurant | no | 15.50 |
| 2025-07-01 19:15:05 | Miami Community Center | fitness | yes | 50.00 |
| 2025-07-02 09:00:00 | NBA League Pass | subscription | no | 89.50 |
| 2025-07-03 17:30:45 | Publix Super Markets | grocery | yes | 122.31 |
| 2025-07-03 19:10:12 | Spotify Premium | subscription | no | 9.99 |
| 2025-07-04 18:20:58 | Publix Super Markets | grocery | yes | 158.50 |
| 2025-07-04 20:05:21 | Shell | gas | yes | 62.31 |
| 2025-07-05 10:00:00 | payment thank you | payment | no | -1500.00 |
| 2025-07-05 19:00:15 | Publix Super Markets | grocery | yes | 185.51 |
| 2025-07-06 11:00:00 | Miami Museum of Art | subscription | no | 75.00 |
| 2025-07-06 17:40:02 | Publix Super Markets | grocery | yes | 109.99 |
| 2025-07-06 19:10:44 | Miami Community Center | fitness | yes | 50.00 |
| 2025-07-07 13:05:19 | Dick's Sporting Goods | retail | yes | 95.80 |
| 2025-07-08 10:00:00 | Art Supply Warehouse | subscription | no | 48.99 |
| 2025-07-08 19:20:50 | Casa Tua | restaurant | yes | 215.45 |
| 2025-07-09 08:00:15 | La Colombe Coffee Roasters | cafe | yes | 4.75 |
| 2025-07-09 18:45:02 | Parking Garage | parking | yes | 10.00 |
| 2025-07-09 19:10:45 | Publix Super Markets | grocery | yes | 112.31 |

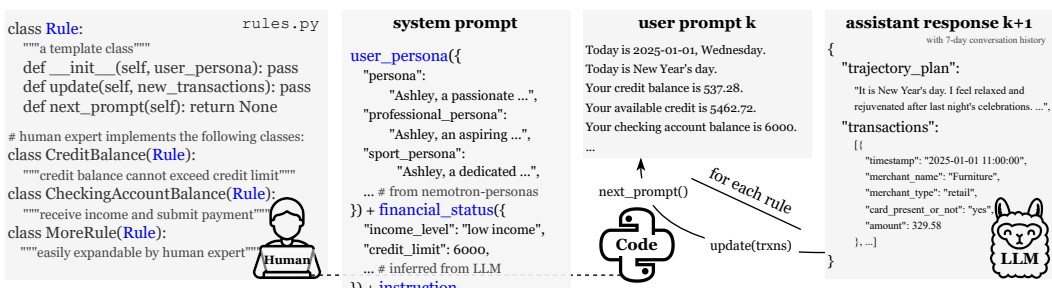

Figure 6: Iterative generation pipeline. A stateful program enforces expert accounting rules and invariants, prompts the LLM for a daily plan, and then updates state for the next cycle.

**Our rule grounded, accounting correct pipeline.** Figure 6 couples an LLM with a programmatic engine that maintains a canonical ledger state and enforces accounting invariants. Human experts encode rules in a simple template with `update` and `next_prompt`. The system prompt is initialized with a persona and derived financial status; the pipeline then proceeds day by day. At each step, a Python program: *(i)* assembles a rule and state aware prompt from the current ledger and a seven day conversation history; *(ii)* receives a daily plan with proposed transactions from the LLM; *(iii)* verifies post conditions and reconciles balances; if any check fails, `next_prompt` returns targeted feedback and requested corrections rather than silently accepting infeasible events; otherwise, `update` applies deterministic state transitions and advances the day. The LLM supplies compositional diversity; the engine guarantees accounting correctness.

**Human-implemented rules to ensure accounting correctness and invariants.** To offload accounting and scheduling from the LLM, a set of rules and programmatic invariants enforce these constraints before accepting any LLM proposal, ensuring that all state transitions are valid:

- **Cash conservation.** $\text{cash}_{t+1} = \text{cash}_t + \text{income}_t - \text{payment}_t$, ensuring no silent creation of funds.
- **Credit balance.** $\text{bal}_{t+1} = \text{bal}_t + \text{spending}_t - \text{payment}_t + \text{interest}_t + \text{fees}_t$, with $0 \leq \text{bal}_{t+1} \leq \text{limit}$.
- **Due date compliance.** Payments must occur on or before due dates; otherwise `next_prompt` requires remediation (late fee, interest) in the next plan.
- **Subscription Carryover.** Carries forward and charges recurring subscriptions on schedule. Active subscriptions must appear each billing period until an explicit cancel or pause event is generated.
- **Liquidity and Solvency.** Over user-defined windows, plans are rejected with a liquidity warning if $\sum \text{outflows} > \sum \text{inflows} + \text{starting\_cash} + \text{available\_credit}$. The simulation terminates if the checking balance falls below the allowed overdraft, marking the user as illiquid.

**Human-implemented rules to mimic real-world scenarios.** These rules act as expandable programmatic guardrails, introducing plausible constraints and variability into the simulation.

- **Temporal and Purchase Cadence.** Provides calendar context (date, day, holiday) to induce time-based patterns and tracks last purchase dates (e.g., groceries, fuel) to maintain realistic frequencies.
- **Random Events.** simulate occasional large expenses by triggering a predefined random event (e.g., appliance failure) with a 10% daily probability.

All checks and updates are logged; every accepted transaction has a matching state transition, enabling full audit and exact regeneration.

**User personas and financial status.** We initialize the system with a persona drawn from the Nemotron Personas collection (Meyer & Corneil, 2025) and a derived financial status. A Llama-3.3-70B model infers the status from the persona using seven expert crafted in context examples. The status includes income level, credit limit, payment habits, subscriptions, recurring bills, car ownership, and spending patterns. Most fields enter the system prompt, while subscriptions and recurring bills are executed by the program to ensure correct carryover. See Appendix A.

## 3 BENCHMARKING

### 3.1 BENCHMARKING TASKS

**Two complementary tasks that are realistic, difficult, and designed not to saturate.** From raw event sequences we derive one *user-level* and one *event-level* task. **Illiquidity classification:** given a user's $n$ month history, predict whether the user will become illiquid (credit card balance exceeding available cash flow at specific moments). **Identity theft segmentation:** to simulate stolen card behavior, we inject one day of chronologically aligned transactions from a secondary user into a primary user's $n$ month history; the goal is to label each transaction as fraudulent (secondary user) or legitimate (primary user). We create 150,000 training and 36,000 testing sequences for each task.

**A single difficulty knob controls headroom and prevents saturation.** We vary the $n$ *month context* to tune challenge. For illiquidity classification, longer context reduces difficulty by revealing more of the user's rhythm. For identity theft segmentation, longer context raises difficulty by enlarging the normal baseline and making the injected day harder to isolate. With $n=3$, sequences average 163 events, the positive rate is $3.43\%$ for illiquidity and $1.13\%$ for identity theft. Because $n$ is unbounded and can be paired with richer personas, new rules, and shifted economic conditions, the suite scales in difficulty and will not become obsolete as models improve.

**Tasks capture real financial structure rather than label shortcuts.** Illiquidity is an *emergent property* of the simulation, not a pre assigned tag, arising when persona driven spending outpaces simulated income. Fraud events are *behavioral overlays* from another real user, not simply random anomalies sampled from a different statistical distribution, forcing models to learn a user-specific baseline and detect coherent yet misplaced activity.

**Distinct capabilities are tested by design.** Illiquidity classification probes early warning from weak signals and rare events that foreshadow liquidity stress. Identity theft segmentation probes fine-grained deviation detection under class imbalance where normal and fraudulent purchases may share merchant types and amounts.

## 3.2 EVALUATION SETTING

**Strong baselines under controlled protocols.** We evaluate state of the art time series models using implementations adapted directly from Wang et al. (2024). For each task we report results at two difficulty settings, $n=1$ and $n=3$.

**Training and hyperparameters.** All models are trained on a single NVIDIA A10G GPU with 24 GB memory. Batch sizes maximize device utilization. We use Adam and select the learning rate from 1e-4, 5e-4, 1e-3 based on validation performance, reporting the best for each model.

**Loss functions and imbalance handling.** For illiquidity classification we balance the training sampler to present equal positive and negative users per batch. For identity theft segmentation we use binary cross entropy with positive class weight equal to the inverse of its batch frequency.

**Metrics that reflect both thresholded decisions and ranking quality.** We report F1 and AUC scores. F1 is computed at the probability threshold that maximizes validation F1 to account for imbalance; AUC provides a threshold independent ranking measure where $0.5$ is random.

**Input representation respects signs and long tails.** Transaction amounts can be positive or negative and follow a long tailed distribution. We preserve sign while compressing magnitude by:

$$sgn\_log\_amount = \text{sign}(amount) * \log(1 + |amount|) \tag{1}$$

Merchant name and merchant type are encoded with frequency-based one-hot features: categories with more than 5,000 occurrences receive unique indicators, others map to an "unknown" bucket. This yields 418 dimensions for merchant names and 266 for merchant types. Together with card_present_or_not and sgn_log_amount, the final input dimension is 686.

**Why the suite will not saturate.** Beyond the $n$ month knob, our programmatic rules and persona library allow systematic stress testing: tighter credit limits, altered pay cadence, new subscription patterns, economic shocks to prices or employment, and calendar regimes with denser holiday effects. Because groundedness is enforced by the engine, these changes remain consistent with accounting, enabling principled difficulty escalation and sustained headroom without retraining the generator.

## 3.3 EVALUATION DISCUSSION

The experimental results, presented in Tables 2 and 3, provide several key insights into model performance and the intrinsic difficulty of the tasks.

**Performance Disparity Between Tasks.** There was a notable difference in model performance between the two tasks. For illiquidity classification, all models demonstrated an ability to learn meaningful features, achieving AUC scores well above the 0.5 random baseline. However, identity theft segmentation proved to be substantially more difficult, with several models struggling to perform better than random chance.

**The Surprising Effectiveness of a Standard Transformer.** Interestingly, a standard Transformer model achieved highly competitive performance on both tasks. This suggests that financial transaction sequences may not exhibit the strong temporal dependencies (e.g., seasonality, auto-correlation) typical of traditional time-series data. As a result, the specialized architectures of newer models may provide limited advantages, whereas the Transformer's general and powerful sequence-to-sequence attention mechanism remains highly effective.

**Effect of $n$-month context.** The $n$-month context serves effectively as the difficulty knob. The performances of all models are varied consistently when the context length changes.

**Low F1 Scores and Task Challenges.** The relatively low F1 scores, even for top-performing models, underscore the fundamental challenges inherent in these tasks. For illiquidity classification, a user's illiquidity can be triggered by sudden, unpredictable events (e.g., a large emergency purchase) that do not follow discernible historical patterns, making them inherently difficult to predict. For identity theft: the boundary between legitimate and fraudulent behavior is often ambiguous. Many common transactions, like grocery shopping or gas refills, are shared between a user's normal activity and fraudulent patterns, making them fundamentally hard to distinguish.

Table 2: Performance of illiquidity classification. A green score indicates a better performance.

| Illiquidity Prediction | 3-month context length | | | | 1-month context length | | | |
|---|---|---|---|---|---|---|---|---|
| | Precision | Recall | F1 ↑ | AUC ↑ | Precision | Recall | F1 ↑ | AUC ↑ |
| MICN (Wang et al., 2023) | 0.085 | 0.514 | 0.145 | 0.718 | 0.087 | 0.657 | 0.153 | 0.755 |
| LightTS (Zhang et al.) | 0.107 | 0.395 | 0.167 | 0.725 | 0.093 | 0.610 | 0.161 | 0.746 |
| DLinear (Zeng et al., 2023) | 0.103 | 0.473 | 0.168 | 0.735 | 0.098 | 0.626 | 0.169 | 0.758 |
| FiLM (Zhou et al., 2022a) | 0.115 | 0.516 | 0.188 | 0.762 | 0.112 | 0.595 | 0.189 | 0.774 |
| Autoformer (Wu et al., 2021) | 0.105 | 0.543 | 0.175 | 0.763 | 0.099 | 0.653 | 0.172 | 0.773 |
| iTransformer (Liu et al., 2023) | 0.126 | 0.530 | 0.202 | 0.775 | 0.103 | 0.655 | 0.178 | 0.785 |
| FEDformer (Zhou et al., 2022b) | 0.083 | 0.733 | 0.149 | 0.791 | 0.099 | 0.649 | 0.172 | 0.776 |
| Crossformer (Zhang & Yan, 2023) | 0.107 | 0.645 | 0.184 | 0.795 | 0.097 | 0.702 | 0.169 | 0.789 |
| Informer (Zhou et al., 2021) | 0.120 | 0.632 | 0.199 | 0.804 | 0.117 | 0.609 | 0.197 | 0.784 |
| ETSformer (Woo et al., 2022) | 0.127 | 0.634 | 0.207 | 0.811 | 0.104 | 0.673 | 0.179 | 0.789 |
| TimesNet (Wu et al., 2022) | 0.101 | 0.727 | 0.175 | 0.816 | 0.126 | 0.589 | 0.208 | 0.797 |
| Transformer (Vaswani et al., 2017) | 0.122 | 0.668 | 0.204 | 0.817 | 0.097 | 0.658 | 0.169 | 0.774 |
| Reformer (Kitaev et al., 2020) | 0.118 | 0.690 | 0.195 | 0.819 | 0.107 | 0.662 | 0.185 | 0.794 |
| PatchTST (Nie et al., 2022) | 0.129 | 0.660 | 0.216 | 0.823 | 0.104 | 0.687 | 0.180 | 0.795 |
| Pyraformer (Liu et al., 2022b) | 0.098 | 0.767 | 0.174 | 0.828 | 0.108 | 0.696 | 0.186 | 0.804 |

Table 3: Performance of identity theft segmentation. A green score indicates a better performance.

| Identity Theft Segmentation | 3-month context length | | | | 1-month context length | | | |
|---|---|---|---|---|---|---|---|---|
| | Precision | Recall | F1 ↑ | AUC ↑ | Precision | Recall | F1 ↑ | AUC ↑ |
| Crossformer (Zhang & Yan, 2023) | 0.033 | 0.326 | 0.061 | 0.518 | 0.034 | 0.327 | 0.061 | 0.519 |
| DLinear (Zeng et al., 2023) | 0.034 | 0.313 | 0.061 | 0.518 | 0.034 | 0.338 | 0.061 | 0.520 |
| FiLM (Zhou et al., 2022a) | 0.034 | 0.308 | 0.061 | 0.519 | 0.031 | 1.000 | 0.060 | 0.503 |
| iTransformer (Liu et al., 2023) | 0.034 | 0.313 | 0.061 | 0.520 | 0.035 | 0.531 | 0.065 | 0.551 |
| FEDformer (Zhou et al., 2022b) | 0.050 | 0.054 | 0.052 | 0.607 | 0.342 | 0.329 | 0.335 | 0.783 |
| ETSformer (Woo et al., 2022) | 0.025 | 0.169 | 0.044 | 0.611 | 0.369 | 0.272 | 0.313 | 0.759 |
| LightTS (Zhang et al.) | 0.024 | 0.171 | 0.041 | 0.618 | 0.293 | 0.379 | 0.330 | 0.806 |
| Autoformer (Wu et al., 2021) | 0.314 | 0.100 | 0.151 | 0.711 | 0.553 | 0.308 | 0.395 | 0.797 |
| MICN (Wang et al., 2023) | 0.314 | 0.152 | 0.205 | 0.727 | 0.544 | 0.323 | 0.405 | 0.827 |
| Reformer (Kitaev et al., 2020) | 0.384 | 0.104 | 0.164 | 0.730 | 0.561 | 0.376 | 0.451 | 0.843 |
| Pyraformer (Liu et al., 2022b) | 0.277 | 0.152 | 0.197 | 0.743 | 0.534 | 0.347 | 0.421 | 0.847 |
| Informer (Zhou et al., 2021) | 0.149 | 0.231 | 0.181 | 0.752 | 0.557 | 0.351 | 0.430 | 0.832 |
| Transformer (Vaswani et al., 2017) | 0.326 | 0.219 | 0.262 | 0.772 | 0.574 | 0.406 | 0.476 | 0.870 |
| TimesNet (Wu et al., 2022) | 0.105 | 0.571 | 0.177 | 0.790 | 0.392 | 0.375 | 0.383 | 0.831 |

## 3.4 POTENTIAL MODEL IMPROVEMENT

While our current benchmark uses straightforward feature engineering, future research could unlock better performance by leveraging more advanced techniques. Promising directions include developing sophisticated **timestamp encodings**, such as learnable time embeddings, to capture both absolute and relative temporal dynamics. Another approach involves applying Natural Language Processing to create dense **semantic embeddings for merchants**, which would capture inherent relationships between them and provide richer representations for infrequent categories. Furthermore, the availability of massive unlabeled transaction sequences is ideal for **self-supervised pre-training**, where a large model could first learn general user spending behaviors before being fine-tuned for specific downstream tasks like identity theft detection and illiquidity classification.

## 4 RELATED WORK

### 4.1 SYNTHETIC TRANSACTIONS DATASETS

Financial industries generates huge amounts of transaction events every day. However, strict regulations require this data to be kept private for internal use only, which limits academic and public research. To overcome this barrier, researchers are developing and using synthetic transaction datasets, where existing synthetic data generation can be broadly categorized into these two approaches:

**Rule-based approaches** rely on simplistic techniques such as stochastic sampling (Padhi et al., 2021; Altman, 2021) and hand-crafted patterns (Altman et al., 2023). Although these approaches introduced a degree of data diversity, they often failed to capture the complex behavioral and economic patterns in real financial data (Kannan, 2024). **Deep-learning-based approaches** build upon vari-

ational autoencoders (VAEs) and generative adversarial networks (GANs) (Figueira & Vaz, 2022) to inject more diversity to the generation. Representative work includes Carvajal-Patiño & Ramos-Pollán (2022); Liu et al. (2022a); Nickerson et al. (2022); Madhavi et al. (2023); Kannan (2024); Kazadaev et al. (2024). While the generated data may be diverse, it only reflects the statistical patterns of the training data and is not grounded in real-world constraints. More recently, the emergence of Large Language Models (LLMs) has created significant potential for generating synthetic financial data. For example, Zuo et al. (2024) used a policy network to select prompts and an LLM to execute them, producing synthetic data for sentiment analysis, while using LLMs for generating transactions has not been explored.

In this work, we explore using LLMs to generate transactions sequences. Our approach is designed to overcome the limitations of both rule-based and deep-learning approaches by producing data that is both diverse and compliant with real-world rules.

## 4.2 Codes × Large Language Models

Chain-of-thought (CoT) prompting (Wei et al., 2022) is an effective technique where language models produce text describing their reasoning, computation, and final answer to a question. However, Large Language Models (LLMs) often struggle with mathematical and numerical tasks where precision is critical. To overcome this weakness, many approaches combine LLMs with the reliable and low-cost execution of code. These methods generally fall into three main categories: **LLM as Codex:** The LLM is prompted to generate executable code that solves a specific task (Chen et al., 2022). **LLM uses tools:** The LLM is given access to external tools or APIs. It queries them, and the responses are fed back to the model as new context (Qu et al., 2025). **Code as verifier:** The LLM's output is passed to a program that verifies its correctness (Ni et al., 2023). Our method jointly belongs to "LLM uses tools" and "Code as verifier", where the state information is computed by the user-defined rules, and the user's illiquidity status is verified by the program.

## 4.3 LLM-Based Simulation

By mimicking human reasoning, Large Language Models (LLMs) allow researchers to simulate human behavior and study complex social and economic patterns. For example, LLM-based simulation is used to create virtual personas that align with real-world demographic distributions (Inoshita & Harada, 2025; Meyer & Corneil, 2025); to study user behavior in healthcare (Li et al., 2024), education (Gao et al., 2025), and social interaction (Wang et al., 2025; Yang et al., 2024; Tang et al., 2024); to help refine recommendation systems (Ma et al., 2025; Ebrat et al., 2024; Gu et al., 2025; Zhang et al., 2025; Ye et al., 2025) and advance economic research in areas such as game theory (Orlando et al., 2025), macroeconomics (Dwarakanath et al., 2024b; Brusatin et al., 2024; Dwarakanath et al., 2024a), and finance (Batra et al., 2025). Our work is another instance of LLM-based simulation, where we simulate the spending and payment behavior of credit card users.

## 5 Conclusion

This work addresses the critical shortage of public, high-fidelity financial transaction data for academic research. We resolve this issue with PersonaLedger, a novel data generation engine that combines the behavioral diversity of Large Language Models with the logical consistency of a programmatic, rule-based financial simulator. Using this engine, we created and publicly released a large-scale benchmark dataset containing 30 million transactions from 23,000 distinct user personas. To demonstrate its utility and establish a performance baseline, we evaluated a suite of state-of-the-art time series models on two challenging downstream tasks: illiquidity classification and identity theft segmentation. By providing a rich, realistic, and privacy-preserving resource, our dataset provides a foundational platform for the research community. It enables the development, validation, and standardized comparison of new methodologies for analyzing financial transaction data, thereby accelerating innovation in the field. Future work includes expanding the rule set to further improve the realism, such as grocery and gas inventory tracking, travel simulation, and price modeling. We also plan to further scale up the dataset size by inputting more user personas.

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

## A  LIST OF FIELDS AND EXAMPLES

### A.1  AUGMENTED USER PERSONA

The following code block provides a complete example of an augmented persona. This persona consists of two parts: a `user_persona`, taken directly from Nemotron-Persona, and a `user_financial_profile`, which is inferred from the `user_persona` by Llama-3 70B. A programmatic module then reads the income level and credit limit from this financial profile to initialize the user's state. As illustrated in Figure 6, the entire augmented persona is injected into the system prompt.

```
{
  "user_persona": {
   "persona": "A disciplined, sociable visionary, Jonathan balances
      practicality with curiosity, leaving a lasting impact on his
      community through his organized, competitive approach",
   "professional_persona": "A retired manufacturing manager, Jonathan now
      excels as a community developer, leveraging his organizational
      skills and competitive nature to drive sustainable growth in
      Wickliffe",
   "sports_persona": "An avid golfer, Jonathan plays weekly at the
      Wickliffe Country Club and cheers for the Cleveland Browns,
      maintaining his competitive spirit even in leisure",
   "arts_persona": "A history enthusiast, Jonathan often leads tours at
      the Lake County Historical Society, sharing stories about local
      pioneers and their impact on the region's development",
   "travel_persona": "A seasoned, meticulous planner, Jonathan favors
      international destinations with rich histories, like Edinburgh and
      Dublin, where he can explore ancestral roots and enjoy a round of
      golf at prestigious courses",
   "culinary_persona": "A fan of hearty, Midwestern comfort food, Jonathan
       enjoys cooking traditional family recipes, like his grandmother's
      beef stew, and hosting potlucks at his home",
   "skills_and_expertise_list": "['project management', 'budgeting and
      financial planning', 'negotiation', 'community development', '
      fundraising']",
   "hobbies_and_interests_list": "['golfing', 'woodworking', 'coin
      collecting', 'history', 'board games and puzzles']",
   "career_goals_and_ambitions": "After retiring from his career in
      manufacturing management, Jonathan has focused his ambition on
      community development. He's actively involved in the Wickliffe
      Chamber of Commerce, aiming to bring new businesses to the town. He
       also serves on the Lake County Planning Commission, working
      towards sustainable development. Despite his competitive nature, he
      's more interested in leaving a lasting impact on his community
      than personal gain.",
   "sex": "Male",
   "age": "72",
   "marital_status": "widowed",
   "education_level": "high_school",
   "bachelors_field": null,
   "occupation": "not_in_workforce"
  },
  "user_financial_profile": {
   "income_level": "med income",
   "credit_limit": 9500,
   "payment_habit": "automatic_payment",
   "car_ownership": "owns_1_car",
   "spending_patterns": "Balancers: intentionally prioritize saving and
      investing for the future while still maintaining a comfortable
      current lifestyle."
  }
}
```

## A.2 HANDLING SUBSCRIPTIONS AND RECURRING BILLS

In addition to basic financial information, we also infer the user's subscriptions (fixed, regular payments) and recurring bills (variable, regular payments). Unlike the core financial profile, this information is not passed to the system prompt but is handled directly by a programmatic module. To keep the user informed, the engine automatically sends notifications for these charges on their scheduled payment dates.

```
"subscriptions": [
  {
    "date_to_charge": 5,
    "amount": 12.99,
    "charge_frequency_month": 1,
    "std": 0.0,
    "merchant_name": "PGA Tour+",
    "product_description": "Golf Streaming Service"
  },
  {
    "date_to_charge": 10,
    "amount": 15.0,
    "charge_frequency_month": 1,
    "std": 0.0,
    "merchant_name": "History Channel",
    "product_description": "History Programming Subscription"
  },
  {
    "date_to_charge": 15,
    "amount": 30.0,
    "charge_frequency_month": 3,
    "std": 0.0,
    "merchant_name": "Woodcraft Magazine",
    "product_description": "Woodworking Magazine"
  },
  {
    "date_to_charge": 20,
    "amount": 25.0,
    "charge_frequency_month": 1,
    "std": 0.0,
    "merchant_name": "Numismatic News",
    "product_description": "Coin Collecting Magazine"
  },
  {
    "date_to_charge": 25,
    "amount": 15.49,
    "charge_frequency_month": 1,
    "std": 0.0,
    "merchant_name": "Netflix",
    "product_description": "Streaming Service"
  }
],
"recurring_variable_bills": [
  {
    "date_to_charge": 1,
    "amount": 120.0,
    "charge_frequency_month": 1,
    "std": 30.0,
    "merchant_name": "FirstEnergy",
    "product_description": "Electricity Bill"
  },
  {
    "date_to_charge": 5,
    "amount": 45.0,
    "charge_frequency_month": 1,
    "std": 10.0,
    "merchant_name": "City Water Department",
```

```
 58      "product_description": "Water Bill"
 59     },
 60     {
 61      "date_to_charge": 10,
 62      "amount": 800.0,
 63      "charge_frequency_month": 6,
 64      "std": 20.0,
 65      "merchant_name": "State Farm",
 66      "product_description": "Homeowner's Insurance"
 67     },
 68     {
 69      "date_to_charge": 1,
 70      "amount": 1500.0,
 71      "charge_frequency_month": 12,
 72      "std": 0.0,
 73      "merchant_name": "Lake County Tax Assessor",
 74      "product_description": "Property Tax"
 75     },
 76     {
 77      "date_to_charge": 15,
 78      "amount": 70.0,
 79      "charge_frequency_month": 1,
 80      "std": 5.0,
 81      "merchant_name": "Xfinity",
 82      "product_description": "Internet Service"
 83     }
 84    ]
```

## B    EXPLORATORY DATA ANALYSIS

Figure 7 plots the credit utilization rate (cumulative balance divided by credit limit) over one year. The data reveals that illiquid users tend to have more erratic spending and payment behaviors, resulting in a significantly more volatile credit utilization curve compared to that of typical users. Figure 8 presents a word cloud of the merchant names and types, highlighting the diversity of merchants contained in our dataset.

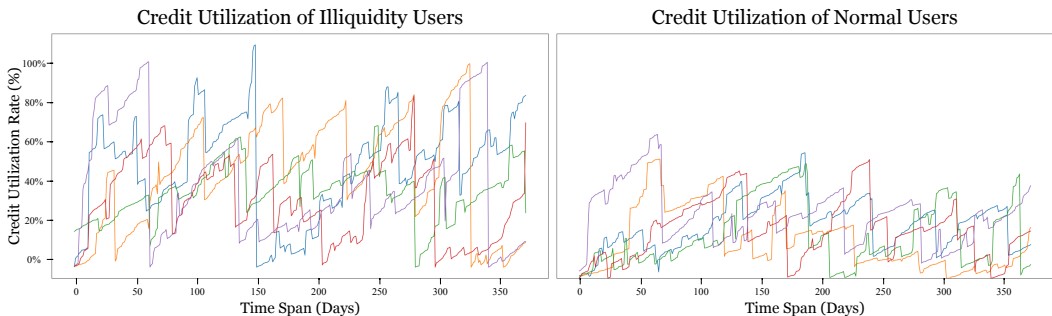

Figure 7: A comparison of credit utilization rates between solvent and illiquid users. Illiquid users display a significantly more volatile credit utilization curve than typical users.

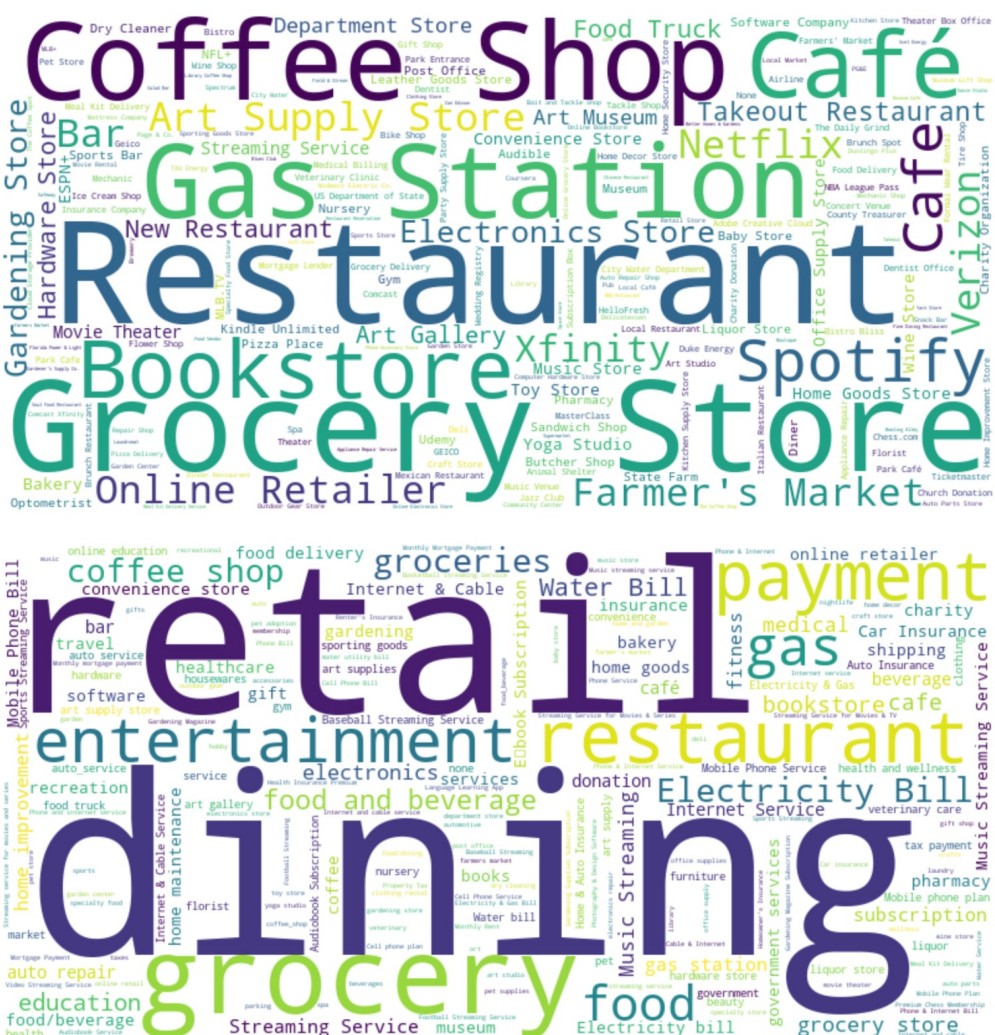

Figure 8: Word cloud visualization of merchant names (upper) and merchant types (lower).

