# OpenReview forum: "PersonaLedger: Generating Realistic Financial Transactions with Persona Conditioned LLMs and Rule Grounded Feedback"
_ICLR.cc/2026/Conference — Submitted to ICLR 2026_

### Official Review · Reviewer_p5Pj · 2025-10-30

**Soundness:** 2
**Presentation:** 3
**Contribution:** 2
**Rating:** 4
**Confidence:** 4

**Summary:**

The paper introduces PersonaLedger, a system designed to generate realistic financial transaction data while preserving privacy. It combines persona-conditioned large language models (LLMs) with a programmatic rule-driven engine to create transaction streams that are both diverse and compliant with real-world financial constraints. This framework allows for the generation of 30 million transactions from 23,000 users, providing a synthetic dataset that can be used for financial forecasting and anomaly detection tasks such as illiquidity classification and identity theft segmentation. The system also ensures that the generated transactions respect hard financial rules, such as income-to-spending ratios and timely payments.

**Strengths:**

Originality: The paper introduces a new approach for generating synthetic financial data by combining LLMs with a rule-based system, which ensures both diversity and logical consistency. This is an innovative step forward in financial AI research, where previous methods have struggled to balance these two aspects.

Quality: The proposed system is well-designed and effectively ensures that financial transactions respect real-world constraints. The dataset is large and diverse, providing a valuable resource for future research in financial anomaly detection and forecasting.

Clarity: The paper is generally well-written, and the figures and tables are effective in illustrating the system’s design and the experimental results. Some sections could benefit from more detailed explanations, but overall, the clarity of the writing is good.

Significance: The contribution of the paper is significant, especially in the context of financial AI research, which often suffers from a lack of public, high-quality datasets. PersonaLedger addresses this gap by offering a publicly available dataset that is both diverse and realistic. Moreover, the introduction of the rule-grounded generation engine is a novel approach to ensuring financial consistency in synthetic data generation. However, the benchmarking tasks and their evaluation could be more rigorously compared against existing benchmarks to strengthen the paper’s claims of originality and impact.

**Weaknesses:**

1 Lack of Comparative Validation: The benchmark tasks are only tested on the PersonaLedger dataset, without validation against real-world financial data or other synthetic datasets, making it unclear if the models would perform similarly in practical scenarios.

2 Insufficient Quantitative Support for Claims: Claims about the dataset's "socioeconomically realistic" patterns lack statistical validation, such as p-values or confidence intervals, weakening their credibility.

3 Unclear Merchant Name Selection: The paper does not explain how the LLM selects specific merchants (e.g., Walmart vs. Kroger), which could impact the dataset’s realism and utility for merchant-specific analysis or fraud detection.

4 Static User Personas: The Personas used in the dataset are static, without temporal evolution or integration of life events (e.g., marriage, retirement), which limits the dataset’s ability to capture long-term or evolving user behaviors.

**Questions:**

1 LLM Degradation Over Time: The paper mentions that LLMs may struggle with rule consistency as transaction sequences grow longer. Could you provide more detailed empirical results on how the system maintains financial accuracy in long sequences, and what steps can be taken to mitigate any degradation in performance over time?

2 Handling Rare Events: How does PersonaLedger handle rare or unusual user behaviors that significantly deviate from typical patterns (e.g., sudden large expenses or life events)? Are these edge cases adequately captured, and how does the system adapt to them?

3 Benchmarking Comparison: How does PersonaLedger compare to other synthetic financial datasets in terms of task difficulty and the realism of generated transactions? Could you provide a more direct comparison with other widely-used datasets for tasks like fraud detection or financial forecasting to validate the uniqueness and value of your approach?

---

> ### Author Response · Authors · 2025-11-21
> **Responses to Weaknesses**
>
> We sincerely appreciate your constructive feedback. Below, we address the specific weaknesses and questions you raised.
>
> > W1: Lack of comparative validation against real-world financial data.
>
> We agree that comparing our results against real data is the ideal method. However, we cannot access real-world data, even on a small scale. Due to strict privacy constraints, publicly available transaction data does not exist.
>
> Despite this limitation, we provide sample transactions for your review. We invite you to inspect them to see if the data aligns with your intuition. Please refer to "**Re: Comparison with Real-World Transaction Data**" and "**Re: Realism – Additional Evaluation and Comparison**" in our general response.
>
> > W2: Claims about the dataset's "socioeconomically realistic" patterns lack statistical validation.
>
> We’d like to politely clarify that our goal is structural realism, not perfect replication of the statistics. Specifically, as detailed in the paper:
>
> * Our agents exhibit expected spending differences between holidays/non-holidays and weekdays/weekends.
> The age distribution follows the well-known lifecycle spending curve.
> * We do not aim to match precise absolute numbers (e.g., the exact average dollar amount spent per age group found in Census Bureau data). Our primary goal is to create data that is structurally sound and useful for training fraud detection systems. We focus on these relative behavioral relationships. We will revise our wording in the paper to use "structural and behavioral realism" for greater clarity.
>
> > W3: Unclear Merchant Name Selection.
>
> The selection of merchant names is entirely handled by the LLM during synthetic data generation. Since the internal workings of LLM text generation are opaque, we rely on the following rationale:
> * Implicit Marketplace Realism: The LLM has been trained on a massive and diverse corpus of public internet data. This vast exposure means the LLM has already learned the relative frequency and context of different merchants within various purchase categories and geographies. For example, it is inherently more likely to generate a transaction at "Walmart" than a hyper-local, obscure retailer.
> * Prompting as Constraint: Our primary method for controlling this selection is through prompting, which constrains the LLM to select merchants appropriate for the specific transaction category (e.g., only generating a grocery merchant when the transaction is for food).
>
> > W4: Static User Personas.
>
> That is a good idea! We agree that incorporating major life events, such as marriage or retirement, would significantly enhance the realism of our user personas. This is entirely feasible and can be easily integrated by expanding our current human-implemented rule set.

---

> ### Author Response · Authors · 2025-11-21
> **Responses to Questions**
>
> > Q1: LLM Degradation Over Time? How does the system maintain financial accuracy in long sequences?
>
> Our framework guarantees 100% financial accuracy, regardless of the transaction sequence length. This is because the calculation of running and credit balances is handled entirely by the deterministic Pythonic module, not by the LLM itself.
>
> The degradation mentioned in the paper regarding “rule consistency” refers specifically to the direct prompting scenario (without the code module), as shown in Figure 5. Standard LLMs often fail at math over long sequences. Our framework is designed exactly to solve this problem by offloading the computation to the code engine.
>
> > Q2: How does PersonaLedger handle rare or unusual user behaviors that significantly deviate from typical patterns (e.g., sudden large expenses or life events)?
>
> PersonaLedger handles rare events or outliers through a combination of strict financial tracking and contextual memory.
> 1. Financial Impact: When a large or unusual expense occurs, the Pythonic module immediately reflects this in the running balance. This ensures the financial state remains accurate, regardless of the anomaly.
> 2. Behavioral Adaptation: The transaction remains in the context window for the subsequent $k=7$ rounds. Consequently, the LLM 'sees' both the unusual event and the resulting change in financial status (e.g., a lower balance).
> 3. Reasoning: By leveraging the LLM's inherent reasoning capabilities, the model naturally adjusts future generations. For example, if a user makes a sudden large purchase, the LLM observes the reduced balance in the context window and tends to generate more conservative spending in subsequent turns.
>
> > Q3: How does PersonaLedger compare to other synthetic financial datasets in terms of task difficulty and the realism of generated transactions?
>
> We invite the reviewer to read "**Re: Realism – Additional Evaluation and Comparison**" for a comparison between PersonaLedger and IBM transaction data. In short, existing datasets, such as the IBM transaction data, are designed primarily for fraud detection sequences. In contrast, PersonaLedger is designed to model a complete user profile. Because we include data points like running balances, liquidity, and subscription labels, our dataset supports more complex tasks. This allows researchers to study problems beyond simple pattern matching, such as forecasting a user's future cash flow or predicting service churn.

---

> > ### Comment · Reviewer_p5Pj · 2025-11-27
> >
> > Thanks for the rebuttal. I’m going to stick with reject.
> >
> > The main issue for me is that the paper is about generating realistic financial transactions, but in your response you say you can’t access real-world transaction data. Without any comparison to real distributions (or some kind of credible external validation from a partner who does have access), it’s hard to know whether the outputs are actually realistic vs. just consistent with your handcrafted rules and prompts.
> >
> > Given that gap, I don’t think the current evidence is strong enough to support the paper’s central claim, so I’ll maintain my negative score.

---

> > > ### Author Response · Authors · 2025-12-04
> > > **Thanks for the valuable feedback!**
> > >
> > > We sincerely thank Reviewer p5Pj for the feedback. We fully acknowledge the limitation regarding the lack of comparison against real distributions or external validation. However, we must clarify that this **gap is unbridgeable due to legal constraints** rather than a lack of effort.
> > >
> > > Even if we were to collaborate with a partner possessing access to the raw data, federal law explicitly forbids the disclosure of the specific statistics or distributions required for such a comparison. Consequently, we respectfully argue that this constraint should not be grounds for rejection. If this legal reality were treated as a disqualifying factor, it would create an impossible standard, effectively preventing any research in this domain from ever being published.

---

### Official Review · Reviewer_8asp · 2025-11-01

**Soundness:** 2
**Presentation:** 2
**Contribution:** 3
**Rating:** 4
**Confidence:** 3

**Summary:**

This paper proposes PersonaLedger: using persona-conditioned LLMs to generate candidate transactions, then a programmable rule engine performs state updates and constraint verification, forming a "generate→verify→correct→regenerate" closed loop. Based on this engine, the authors synthesize and promise to release a dataset of approximately 30 million transactions from 23,000 users, providing two evaluation tasks: liquidity stress prediction (user-level) and identity theft segmentation (event-level), reporting Precision/Recall/F1/AUC for various baseline models under a unified protocol. The paper emphasizes reproducibility and auditability (code, rules, prompts, seeds, and generation logs).

**Strengths:**

Excellent work, great creativity and idea, congratulations on realizing it!
Clear methodology, closed loop: LLM ensures diversity, rule engine strictly controls accounting and calendar constraints, errors can be corrected through structured prompts.
Complete resources: Large data scale with comprehensive fields, plus two tasks closely related to risk control/anti-fraud with a unified protocol.
Broad baseline coverage: Compares Transformer, PatchTST, Autoformer, iTransformer, etc. under the same settings, providing a reproducible starting point.

**Weaknesses:**

While the paper provides statistical analysis (Section 2.2) and benchmark tasks (Section 3), there is no systematic evaluation of whether the generated transactions are actually realistic compared to real financial data.

**Questions:**

(1) You state that you will open-source the code and dataset, but I haven't seen any related links yet. If this can be confirmed, I will revise my score. "we create a public dataset of 30 million transactions from 23,000 users and a benchmark suite with two tasks" - I can only evaluate you based on existing materials, not just promises. I can consider revising the score when the dataset and codes are realsed.

(2) Can you release some code to facilitate reproduction for reviewers and readers?

---

> ### Author Response · Authors · 2025-11-21
>
> We sincerely appreciate your constructive feedback. Below, we address the specific weaknesses and questions you raised.
>
> > W1: Lacks evaluation against real data.
>
> We agree that comparing our results against real data is the ideal method. However, we cannot access real-world data, even on a small scale. Due to strict privacy constraints, publicly available transaction data does not exist.
>
> Despite this limitation, we provide sample transactions for your review. We invite you to inspect them to see if the data aligns with your intuition. Please refer to "**Re: Comparison with Real-World Transaction Data**" and "**Re: Realism – Additional Evaluation and Comparison**" in our general response.
>
> > Q: Open-source availability and reproducibility.
>
> We are committed to full transparency. Please refer to the "**Strengths and Reproducibility**" section of our response. We have provided our code and data examples there.

---

### Official Review · Reviewer_NDhW · 2025-11-02

**Soundness:** 3
**Presentation:** 2
**Contribution:** 2
**Rating:** 4
**Confidence:** 4

**Summary:**

This paper proposes PersonaLedger, a synthetic data generation engine that combines a persona-conditioned LLM with a rule-based programmatic controller to simulate realistic financial transaction sequences. The closed-loop design ensures that each generated transaction satisfies accounting constraints (e.g., credit limits, cash balances, due dates), while the LLM provides behavioral diversity conditioned on rich user personas derived from Nemotron-Personas. Using this system, the authors generate 30 million transactions for 23,000 users, and release two benchmark tasks: illiquidity classification and identity theft segmentation.

**Strengths:**

1. Timely contribution: The lack of publicly available transaction data due to privacy restrictions is a real bottleneck in financial AI research. The proposed approach represents an ambitious and creative attempt to overcome this constraint while maintaining logical consistency.

2. Interesting idea: The LLM + rule engine closed loop is very interesting. It directly addresses the brittleness of rule-based simulators and the constraint violations of purely generative models (e.g., GANs or VAEs).

3. Transparency and reproducibility: The release plan, including prompts, rules, seeds, and logs, is commendable.

**Weaknesses:**

1. Insufficient validation of realism: The main limitation lies in the lack of quantitative or external validation demonstrating that the generated data are truly realistic or useful proxies for real-world ledgers. Statistical diversity and rule adherence are necessary but not sufficient. Without comparison to real transaction datasets (even if at an aggregated or stylized level), it is hard to judge whether the synthetic data exhibit realistic interdependencies or temporal dynamics.

2. Shallow persona grounding: The process by which financial profiles are inferred from personas (using a Llama-3.3-70B model with only seven expert-crafted examples) appears subjective and fragile. If the inferred income or credit limit deviates from plausible values, the downstream simulation could produce misleading liquidity patterns. The framework assumes personas are rich and realistic, but there is no ablation or validation showing that persona quality drives realistic behavior.

3. Task design and representativeness: The two benchmark tasks, while well-defined, feel somewhat artificial and detached from genuine financial modeling challenges. For instance, “identity theft segmentation” is simulated simply by inserting another user’s transactions into a sequence — a simplistic proxy that may not reflect actual fraud dynamics. Similarly, “illiquidity classification” is driven by the system’s own liquidity rules, creating a self-referential task that may not generalize beyond this synthetic world.

**Questions:**

Can practitioners or researchers genuinely trust and adopt this dataset as a benchmark? Without external calibration or interpretability analysis (e.g., correlation structures, spending autocorrelation, realistic merchant co-occurrence), does the dataset risk becoming a closed synthetic ecosystem—internally consistent but not empirically grounded?

---

> ### Author Response · Authors · 2025-11-21
>
> We sincerely appreciate your constructive feedback. Below, we address the specific weaknesses and questions you raised.
>
> > W1: Insufficient validation of realism.
>
> We understand your primary concern is whether our synthetic transactions accurately reflect real-world scenarios. We agree that comparing our results against real data is the ideal evaluation method. However, we cannot access real-world data, even on a small scale. Due to strict privacy constraints, publicly available transaction data does not exist.
>
> Despite this limitation, we provide sample transactions for your review. We invite you to inspect them to see if the data aligns with your intuition. Please refer to "**Re: Comparison with Real-World Transaction Data**" and "**Re: Realism – Additional Evaluation and Comparison**" in our general response.
>
> > W2.1: Using a Llama-3.3-70B model with only seven expert-crafted examples appears subjective and fragile.
>
> 1. We limited the input to seven examples due to the context constraints of Llama-3.3-70B. As exemplified [here](https://anonymous.4open.science/r/PersonaLedger-3E5D/generation/Generators/fewshot_examples/0000.json), a single example requires about 2,500 tokens. When we added more examples, the LLM had trouble dealing with long contexts. To address this, we ensured the in-context examples to be diverse. We also refined the chain-of-thought reasoning to ensure the output remains grounded.
>
> 2. We do not intend to make precise financial predictions. We only aim to provide logical estimates to initialize the generation, such as students typically earning less than workers. Llama-3.3-70B with seven in-context examples achieves this effectively.
>
> > W2.2: If the inferred income or credit limit deviates from plausible values, the downstream simulation could produce misleading liquidity patterns.
>
> 1. We ensure the values remain realistic by using strict caps. The highest possible credit limit is 15,000, and the highest bi-weekly income is 16,000.
>
> 2. Furthermore, defining "plausible" values is difficult because the real world is highly diverse. Real-world income distributions have a very wide range. The complexity observed in our results reflects this natural financial diversity, not errors in the simulation.
>
> > W2.3: There is no ablation or validation showing that the persona drives realistic behavior.
>
> We respectfully clarify that we have validated how personas drive realistic behavior.
> * Quantitative Validation: As shown in Figure 3, our statistics confirm that different personas result in distinct spending distributions.
> * Qualitative Validation: To see this realism more clearly, we invite you to compare the generated outputs for two specific profiles:
>     * A [retiree](https://anonymous.4open.science/r/PersonaLedger-3E5D/generation/generated_transactions/000000/000000.json) who focuses on golf and technology. He shows financial discipline by frequently paying off credit card balances.
>     * A [full-time worker](https://anonymous.4open.science/r/PersonaLedger-3E5D/generation/generated_transactions/000000/000001.json) who prioritizes lifestyle experiences. He focuses on fine dining, the arts, and pet care.
>
> > W3: The two benchmark tasks, while well-defined, feel somewhat artificial and detached from genuine financial modeling challenges.
>
> That is a very good point. We agree that our benchmark tasks are simplified. They do not replicate the full complexity of real-world finance. However, we chose this simple setting on purpose: to create unambiguous ground truth. This is necessary for a reliable benchmark that can accurately test and compare model performance.
>
> * For illiquidity classification, the labels are unambiguously defined, and it assesses the model’s capability to accurately understand the user's cash flow dynamics.
> * For identity theft segmentation, the labels are unambiguously defined, and it assesses the model’s capability to distinguish between two distinct spending behaviors.
>
> > Q: Without external calibration or interpretability analysis, does the dataset risk becoming a closed synthetic ecosystem?
>
> We acknowledge that, for general public use, the dataset is a closed synthetic ecosystem. However, it maintains a strong structural similarity to the real world. For example, it accurately reflects spending differences between holidays and non-holidays, as well as the standard lifecycle spending curve.
>
> In addition, the system is designed to be adaptable. Institutions with access to proprietary data can easily calibrate this framework to match their specific statistics. Therefore, the current limitation is strictly due to data access, not the capability of the system itself.

---

### Official Review · Reviewer_pyCd · 2025-11-14

**Soundness:** 3
**Presentation:** 4
**Contribution:** 3
**Rating:** 6
**Confidence:** 2

**Summary:**

The authors construct an entirely synthetic financial transaction dataset consisting of 30 million transactions from 23k users. They leverage the world knowledge inside the LLM to generate realistic spending rollouts for a diverse set of profiles which are seeded from nemotron personas.

**Strengths:**

- The framework developed to generate the data looks useful as an artifact. There is thoughtful design on the abstractions in the engine orchestrating the LLM calls. In particular, the interface for adding rules is well thought out.

- The writing of the paper is very good. The authors thoroughly motivate the problem at hand, discuss issues with naive solutions, and give a very clear exposition of the structure and characteristics of the dataset release.

- The data resource is of high quality. A significant amount of manual effort was used to make the data cleaner and more consistent.

**Weaknesses:**

- There is a lack of concrete evaluation criteria proposed to assess the fidelity of the proposed dataset with respect to a real financial transaction dataset. Arguments of realism are mostly qualitative, or pertain to an arbitrarily picked attribute.

- More evidence of the usefulness of the dataset would be appreciated. For the downstream task benchmarks, it would be good to assess whether the synthetic-benchmark-induced rankings of methods align with the ranking of methods on real tasks; alternatively if the synthetic benchmarks can identify systematic gaps in the performance of a model trained on real data; and also if the synthetic data is useful training data for the real task.

**Questions:**

As the data is entirely synthetically generated, it is important to understand to what extent the resource is useful for developing models for real data. Although this appears to be a weak point of the paper, I believe overall the dataset and generation framework are valuable artifacts for the research community and recommend acceptance.

---

> ### Author Response · Authors · 2025-11-21
>
> We sincerely appreciate your constructive feedback. Below, we address the specific weaknesses and questions you raised.
>
> A summary of the raised weaknesses and questions:
> * Weakness 1: Lacks evaluation against **real data**.
> * Weakness 2: Request more evidence of the dataset's usefulness, such as benchmarking, finding model gaps, or serving as training data for **real-world** tasks.
> * Question: How useful is this resource for developing models for **real data**?
>
> We understand your primary concern is whether our synthetic transactions accurately reflect real-world scenarios. We agree that comparing our results against real data is the ideal evaluation method. However, we cannot access real-world data, even on a small scale. Due to strict privacy constraints, publicly available transaction data does not exist.
>
> Despite this limitation, we provide sample transactions for your review. We invite you to inspect them to see if the data aligns with your intuition. Please refer to "**Re: Comparison with Real-World Transaction Data**" and "**Re: Realism – Additional Evaluation and Comparison**" in our general response.

---

### Author Response · Authors · 2025-11-21
**Strengths And Reproducibility**

We sincerely appreciate the high-quality feedback from our reviewers. We are strongly encouraged by the recognition of several key aspects of our work:
* **Dataset Value**: That our dataset is a high-quality, large-scale, and timely contribution that addresses a critical lack of public financial data.
* **Methodology**: That our "LLM + rule engine" loop is an innovative and well-designed solution. We are especially grateful you noted its effectiveness in balancing generative diversity with real-world constraints, a key challenge we aimed to solve.
* **Writing**: That the paper's motivation, structure, and clarity were effective.

**Reproducibility**. We provide the full implementation of PersonaLedger, including all generation and evaluation code, in our anonymous GitHub repository: [anonymous GitHub link](https://anonymous.4open.science/r/PersonaLedger-3E5D/README.md). It includes:
* Clear documentation for the Python environment and dependencies.
* Step-by-step instructions to reproduce the data [generation](https://anonymous.4open.science/r/PersonaLedger-3E5D/generation/README.md) and [benchmarking](https://anonymous.4open.science/r/PersonaLedger-3E5D/evaluation/README.md).
* Example inputs of 2000 user personas and their synthetic financial profiles: [Link](https://anonymous.4open.science/r/PersonaLedger-3E5D/generation/personas/augmented_persona/000000.json).
* Example generation outputs showing transaction sequences and activity traces for four users: [link to outputs](https://anonymous.4open.science/r/PersonaLedger-3E5D/generation/generated_transactions/000000/000000.json).

The entire dataset (23,000 users and 30 million transactions) is generated with simple parallelization. Due to storage limitations, we uploaded the generated transactions from 1500 users [batch1](https://anonymous.4open.science/r/PersonaLedger-3E5D/sample_data/batch1.csv), [batch2](https://anonymous.4open.science/r/PersonaLedger-3E5D/sample_data/batch2.csv), [batch3](https://anonymous.4open.science/r/PersonaLedger-3E5D/sample_data/batch3.csv), [batch4](https://anonymous.4open.science/r/PersonaLedger-3E5D/sample_data/batch4.csv), [batch5](https://anonymous.4open.science/r/PersonaLedger-3E5D/sample_data/batch5.csv),. We will plan to upload the entire dataset to HuggingFace after the review process.

---

### Author Response · Authors · 2025-11-21
**Re: Comparison with Real-World Transaction Data.**

We heartily agree with the reviewers that comparing our synthetic data to real-world transaction data would be an excellent evaluation method. Unfortunately, this comparison is not possible because we cannot access real transaction data, even on a small scale, due to strict privacy and security constraints. In fact, this privacy issue is the main reason that we develop the technique so that the research community can have access to the transaction data.

This barrier highlights a key difference between financial data and other fields. In areas like image generation, large public "ground truth" datasets (like ImageNet) are available for comparison. In finance, no such public ground truth exists. This is not a limitation unique to our work; it is a common challenge for the entire industry:

* The Kaggle "Credit Card Fraud Detection" dataset [1] is based on real data, but all features are obscured with PCA to protect privacy. This makes a true comparison impossible.
* The creators of the IBM synthetic dataset [2, 3] faced the same barrier. Their team was also unable to access real data for comparison and focused instead on presenting detailed statistics.

**The Value of PersonaLedger.**
This limitation does not reduce the value of PersonaLedger. Our dataset is particularly valuable for private organizations (like banks or financial firms) that do have access to their own real data. An organization can inspect the statistical differences between our dataset and its own private data. If the differences are large, they can use our tunable modules to easily adjust the generation process. This allows them to create a large-scale synthetic dataset that closely matches the properties of their private, real-world data.

[1] https://www.kaggle.com/datasets/mlg-ulb/creditcardfraud/data

[2] Padhi, I., Schiff, Y., Melnyk, I., Rigotti, M., Mroueh, Y., Dognin, P., ... & Altman, E. (2021, June). Tabular transformers for modeling multivariate time series. In ICASSP 2021-2021 IEEE International Conference on Acoustics, Speech and Signal Processing (ICASSP) (pp. 3565-3569). IEEE.

[3] Altman, E. (2021, November). Synthesizing credit card transactions. In Proceedings of the Second ACM International Conference on AI in Finance (pp. 1-9).

---

### Author Response · Authors · 2025-11-21
**Re: Realism – Additional Evaluation and Comparison**

We appreciate the reviewers' feedback regarding the lack of in-depth realism evaluation. To address this, we discussed the fundamental difference between our PersonaLedger and the IBM synthetic transaction dataset, which is the largest available.

**Comparison with IBM Synthetic Transaction Dataset.**

First, we present short slices from both dataset:

IBM:

| User | Card | Year | Month | Day | Time | Amount | Use Chip | Merchant Name | Merchant City | Merchant State | Zip | MCC | Errors? | Is Fraud? |
| :--- | :--- | :--- | :--- | :--- | :--- | :--- | :--- | :--- | :--- | :--- | :--- | :--- | :--- | :--- |
| 0 | 0 | 2002 | 9 | 1 | 06:21 | $134.09 | Swipe Transaction | 3527213246127876953 | La Verne | CA | 91750.0 | 5300 | | No |
| 0 | 0 | 2002 | 9 | 1 | 06:42 | $38.48 | Swipe Transaction | -727612092139916043 | Monterey Park | CA | 91754.0 | 5411 | | No |
| 0 | 0 | 2002 | 9 | 2 | 06:22 | $120.34 | Swipe Transaction | -727612092139916043 | Monterey Park | CA | 91754.0 | 5411 | | No |
| 0 | 0 | 2002 | 9 | 2 | 17:45 | $128.95 | Swipe Transaction | 3414527459579106770 | Monterey Park | CA | 91754.0 | 5651 | | No |
| 0 | 0 | 2002 | 9 | 3 | 06:23 | $104.71 | Swipe Transaction | 5817218446178736267 | La Verne | CA | 91750.0 | 5912 | | No |

PersonaLedger:
| user_id | timestamp | merchant_name | merchant_type | card_present_or_not | amount |
| :--- | :--- | :--- | :--- | :--- | :--- |
| 0 | 2024-04-08 09:00:00 | Speedway | gas_station | yes | 25.92 |
| 0 | 2024-04-08 10:30:00 | Walmart | grocery_store | yes | 68.15 |
| 0 | 2024-04-09 12:00:00 | Wickliffe Country Club | restaurant | yes | 19.98 |
| 0 | 2024-04-09 18:30:00 | Tasty Bites | restaurant | no | 24.02 |
| 0 | 2024-04-10 00:00:00 | History Channel | History Programming Subscription | no | 15.0 |
| 0 | 2024-04-10 13:00:00 | The Coffee Spot | coffee_shop | yes | 11.22 |
| 0 | 2024-04-10 18:00:00 | Snack Attack | convenience_store | yes | 16.42 |

Then, we discuss the fundamental improvement of PersonaLedger over IBM dataset.
| Dimension | IBM Dataset | PersonaLedger |
| :--- | :--- | :--- |
| **Methodology** | **Stochastic Simulation:** Uses state machines and Gaussian sampling. | **Gen AI + Control Loop:** Uses a Persona-Conditioned LLM for proposals coupled with a rule-based engine for validation. |
| **Source of Diversity** | **Manual Coding:** Designers must manually encode logic and transition probabilities. | **Emergent:** Behavior emerges from the LLM's world knowledge of how specific personas act. |
| **Source of Logic** | **Ad-hoc:** Requires ad-hoc logic to ensure variables correlate (e.g., wealth vs. travel). | **Intrinsic:** The LLM automatically handles complex correlations (e.g., a "sociable" person visiting a "sports bar") without manual mapping. |
| **Granularity of Context** | **Coarse States:** Limited to broad states like "Home," "Travel," "Weekday," or "Weekend". | **Fine-Grained Narrative:** Captures daily life context (e.g., "New Year's Day," "feeling motivated", "meeting friends"). |
| **Illiquidity & Payments** | Not considered. | Explicitly models the **lifecycle of bills, interest, and cash flow** to generate realistic illiquidity states. |
| **Tunability & Expansion** | **Difficult:** Requires writing new code/states for every new behavior or merchant type. | **Easy & Scalable:** "Plug and play" expansion by simply changing the text prompt (persona) or adding a Python rule class. |
| **Explainability** | **Low:** Data is generated via math/sampling; no textual rationale exists for why a purchase occurred. | **High:** The LLM generates a "trajectory plan" (reasoning) explaining the user's intent before generating the transaction. |

**Realism Evaluation**. We invite you to review the generated transactions [here](https://anonymous.4open.science/r/PersonaLedger-3E5D/generation/generated_transactions/000000/000000.json). This allows you to verify that the transactions and daily activities align with common sense. Please note that this is a large file. The daily activity logs are located in the second half of the file, so you will need to scroll down to view them.  We also uploaded generated sequences of 1500 users [batch1](https://anonymous.4open.science/r/PersonaLedger-3E5D/sample_data/batch1.csv), [batch2](https://anonymous.4open.science/r/PersonaLedger-3E5D/sample_data/batch2.csv), [batch3](https://anonymous.4open.science/r/PersonaLedger-3E5D/sample_data/batch3.csv), [batch4](https://anonymous.4open.science/r/PersonaLedger-3E5D/sample_data/batch4.csv), [batch5](https://anonymous.4open.science/r/PersonaLedger-3E5D/sample_data/batch5.csv) for your inspection. Feel free to raise any concern about unrealism!

---

### Meta-Review · Area_Chair_fQUA · 2026-01-06

**Summary:**

This submission introduces PersonaLedger, a synthetic financial transaction generator that combines a persona-conditioned LLM with a rule-based controller in a closed loop. The system is well-motivated (privacy blocks access to real ledgers), and the engineering is thoughtful: rules enforce hard accounting constraints while the LLM provides behavioral variety. The scale of the released artifact is also appealing, and the authors provide benchmarks for illiquidity prediction and identity-theft segmentation.

**Reviewer Concerns:**

The main concern point is realism validation: the paper’s central claim is that the generated transactions are realistic, yet there is no credible quantitative grounding against real distributions, and the evidence remains mostly qualitative or self-consistent with the simulator’s own rules. Several reviewers also questioned whether the benchmark tasks are representative of real financial modeling challenges (e.g., the “identity theft” mechanism is simplified), and whether the persona-to-financial-profile grounding is robust. On the plus side, reviewers generally liked the clarity of the paper and the reproducibility intent; the authors also clarified release plans and provided additional samples and code links during rebuttal.

**Reviewer Scores:**

Reviewer pyCd (score 6): likely no change
Reviewer NDhW (score 4): likely no change
Reviewer 8asp (score 4): likely increase by 1
Reviewer p5Pj (score 4): no change

---

### Decision · Program_Chairs · 2026-01-26

Reject